# Revisiting Standard and Novel Therapeutic Approaches in Halitosis: A Review

**DOI:** 10.3390/ijerph191811303

**Published:** 2022-09-08

**Authors:** Catarina Izidoro, João Botelho, Vanessa Machado, Ana Mafalda Reis, Luís Proença, Ricardo Castro Alves, José João Mendes

**Affiliations:** 1Periodontology Department, Egas Moniz Dental Clinic (EMDC), Egas Moniz, CRL, 2829-511 Monte de Caparica, Portugal; 2Clinical Research Unit (CRU), Centro de Investigação Interdisciplinar Egas Moniz (CiiEM), Egas Moniz—Cooperativa de Ensino Superior, CRL, 2829-511 Monte de Caparica, Portugal; 3Instituto de Ciências Biomédicas Abel Salazar, School of Health and Life Sciences, University of Porto, 4099-002 Porto, Portugal; 4Neuroradiology Department, Hospital Pedro Hispano, 4464-513 Matosinhos, Portugal; 5Quantitative Methods for Health Research Unit (MQIS), Centro de Investigação Interdisciplinar Egas Moniz (CiiEM), Egas Moniz, CRL, 2829-511 Monte de Caparica, Portugal

**Keywords:** halitosis, periodontal disease, periodontitis, tongue coating, mouthwashes, probiotics

## Abstract

Halitosis, or bad breath, is an oral health problem characterized by an unpleasant malodor emanating from the oral cavity. This condition can have different origins and causes a negative burden in social interactions, communication and quality of life, and can in uncommon cases be indicative of underlying non-oral non-communicable diseases. Most cases of halitosis are due to inadequate oral hygiene, periodontitis and tongue coating, yet the remaining proportion of cases are due to ear–nose–throat-associated (10%) or gastrointestinal/endocrine (5%) disorders. For this reason, the diagnosis, treatment and clinical management of halitosis often require a multidisciplinary team approach. This comprehensive review revisits the etiology of halitosis as well as standard and novel treatment that may contribute to higher clinical success.

## 1. Introduction

Halitosis is characterized by an unpleasant odor emanating from the mouth, from oral or non-oral sources [1,2]. This malodor mainly results from the presence of odoriferous substances–namedvolatile sulfur compounds (VSCs)–present in the exhaled air as a result of the action of anaerobic oral Gram-negative bacteria (*Bacteroides loescheii*, *Centipedaperiodontii*, *Eikenellacorrodens*, *Treponema denticola*, *Prevotella intermedia*, *Porphyromonasgingivalis*, *Fusobacterium nucleatum*, *Selenomonas*, *Eubacterium*, *Bacteroides forsythus and Tannerellaforsythyia*) on substrates containing sulfur [3,4,5]. Hydrogen sulfide, methyl mercaptan and, to a lesser extent, dimethyl sulfide, represent 90% of the VSCs in bad breath [6,7,8].These VSCs result from bacterial metabolism and its pathways are well described (Figure 1). The understanding and the development of VSCs measurement procedures could be used as clinical levels for the diagnosis of halitosis [9,10].

Epidemiological studies report the prevalence of halitosis is estimated to range between 2.4–78% [3,4], and the American Dental Association outlined near 50% of American adults suffer from oral malodor [11]. The reason for such large variability in the prevalence rates are the large variation in methods used for halitosis assessment, whether the data are based on self-reporting or objective measurements of oral malodor, the geographic location, and the year when the study was developed. With such an elevated prevalence, its etiology and patient-reported outcomes become a topic of interest. Although there is also a “disease mongering” around halitosis, it is known that it has a negative impact on patients’ quality of life, especially in interpersonal relationships [12],and when patients experience social and personal embarrassment they tend to seek treatment by a professional [9].

This review aims to provide a snapshot on halitosis etiology and classification, as well as discussing standard and innovative alternative treatments and clinical management strategies.

## 2. Classification of Halitosis

Although several classifications of halitosis are described in the literature, we considered the classification based on the origin of the VSCs which was established by the International Society for Breath Odor Research (ISBOR) [10,13]. As such, halitosis can be categorizedas genuine halitosis or delusional halitosis (Figure 2):Genuine halitosis is an oral malodor that is noticeable and exceeds the socially acceptable level.Genuine halitosis can be classified into physiological halitosis or pathological halitosis [14]. Pathological halitosis can originate from oral diseases—intra-oral halitosis, (e.g., tongue coating, periodontal infections, odontogenic infections, xerostomia, mucosal lesions) or systemic diseases—extra-oral pathologic halitosis (e.g., respiratory tract infections, gastrointestinal disease, metabolic disorders, endocrine system disorders, medication).

Physiological halitosis, mainly originating from the dorsoposterior region of the tongue, consists of a bad odor originating from putrefaction processes in the oral cavity, without any association with a pathological condition. 

2.Delusional halitosis includes pseudohalitosis and halitophobia.

Pseudohalitosis is a condition in which patients are convinced they have oral malodor, but it is not noticed by others. Complaints usually improve after counseling and simple oral hygiene measures. Halitophobia is the condition in which patients maintain the conviction that they have bad breath even after diagnosis and treatment, without any physical or social evidence suggestive of the presence of halitosis. Also called non-real halitosis, halitophobia is understood by the compulsive idea of suffering from bad breath and irritating others with it [15].

In the following subsections, we revisit halitosis risk factors (Section 2.1) and causes (Section 2.2).

### 2.1. Risk Factors for Halitosis

#### 2.1.1. Behavioral Factors

Among the behavioral factors that increase the risk for developing halitosis are smoking, dietary habits and alcohol consumption [16]. Smoking is linked with increased incidence and severity of halitosis [17,18].This largely stems from the existence of cigarette’s high content of VSCs [19], particles responsible for the malodor. Moreover, cigarette smoke actively alters the subgingival microbial ecosystem balance causing an increase in the absolute numbers of VSC-producing bacteria [20,21]. Furthermore, smoking causes hyposalivation and dry mouth (explained on Section 2.1.2). In what diet concerns, VSCs-enriched foods (such as, garlic, onions, durians and spices), can cause transient unpleasant oral malodor, yet this is often not perceivedas a malodor, and is influenced by people’sacceptable breath smell threshold [16]. Chronic alcohol consumption is another potential risk factor for halitosis as a result of oral and hepatic alcohol oxidation, producing acetaldehyde and other odorous byproducts, or as secondary association hyposalivation and dry mouth [17].

#### 2.1.2. Dry Mouth/Xerostomia

One important characteristic of saliva is its lubricant action that promotes self-cleaning of oral tissues by destroying microorganisms, also aiding in speech, swallowing and neutralizes acids of bacterial origin, thus remineralizing enamel. Moreover, saliva contains desquamated epithelial cells, lysed leukocytes, hemopoietic cells, food debris, and microorganisms [22,23]. Dry mouth is characterized by hyposalivation (in other words, reduced salivary flow), hampering the abovementioned functions, favoring anaerobic bacterial putrefaction of food debris that remain in the oral cavity after meals [18,24]. The lack of salivary flow, with a consequent reduction in antimicrobial activity, promotes the increase of cariogenic microorganisms [23]. Consequently, VSC production increases, the main contributor to halitosis [25]. 

Medication is the most common cause of hyposalivation, namely anticholinergics, antihistaminic, antipsychotics, anxiolytics, antidepressant, antihypertensives, diuretics and narcotics [26]. Chronic mouth breathing, radiotherapy, dehydration, vitamin deficiencies, menopause, emotional disturbances and autoimmune conditions (systemic lupus erythematosus, Sjögren’s syndrome, rheumatoid arthritis and scleroderma) can also reduce salivation, as can systemic illness such as diabetes mellitus (DM), nephritis and thyroid dysfunction. About 25% of the elderly suffer from dry mouth [26]. Furthermore, in geriatric age, oral changes occur that can influence the development of malodor: an increase in salivary pH due to amino acid ingestion and a change in oxygen depletion. These changes stimulate the metabolism of Gram-negative bacteria, responsible for the increased production of VSC [27,28]. Although xerostomia is associated with aging, several studies have shown that salivary gland function is maintained in healthy elderly people. Therefore, dry mouth is probably a condition of systemic or extrinsic origin. 

### 2.2. Causes of Halitosis

#### 2.2.1. Extra-Oral Causes

It is estimated that 10–20% of halitosis has non-oral causes [6,29,30,31]. The origins of extraoral halitosis include sources of respiratory tract infections, gastrointestinal diseases, endocrine and hematologic system disorders [29]. The main VSC associated with extra-oral halitosis is dimethyl sulphoxide whereas the main VSCs contributing to intra-oral halitosis are methyl mercaptan and hydrogen sulphide [29,30]. The most common sources are discussed below according to the system involved.

##### Respiratory System

To distinguish the origin of malodor between oral or nasal origin, the expired air must be separated. The technique described consists of asking the patient to close the lips and exhale through the nose. The examiner will notice if the bad odor is coming from the nose. Then, the patient is asked to pinch the nose and exhale through the mouth [29]. In this way, we can distinguish whether the cause of halitosis is of nasal, nasopharynx or paranasal sinuses origin. Bacterial sinusitis mainly develops from acute viral sinusitis, with Streptococcus pneumonia and *Haemophilus influenza* being the main bacteria involved. When purulent mucus is produced, a typical odor appears. The diagnosis is confirmed using radiological or computed tomography (CT) images, where fading is revealed [32].

In 10% of sinusitis cases, one tooth or several teeth are involved. *Peptostretococcus* spp., *Fusobacterium* spp., *Prevotella* spp. and *Porphyromonas* spp. are the bacteria most frequently involved. Since these bacteria are capable of producing VSCs, a clear association with halitosis is available. In the treatment of acute sinusitis, antibiotics are often used, reducing the prevalence of anaerobic pathogens as well as malodor problem [32].

Chronic sinusitis may be the main cause of bad breath coming from the nose [27,28]. The most common symptoms include post-nasal drip, cough, covered tongue and constant urge to clear the throat [33,34]. It is thought that the purulent material falls on the base of the tongue, which already contains microbiota, predisposing to the production of VSC [35]. Causes of chronic sinusitis can include an upper respiratory tract infection, chronic mucosal disease, or malformed intranasal structures. Patients with halitosis and chronic nasal/paranasal sinus diseases should be referred to an ear-nose-and-throat specialist for causal treatment [35,36].

Repetitive infections of the tonsils and adenoids cause chronic follicular tonsillitis [37]. There are deep crypts that form in the tonsillar crypts which, due to their anatomy, favor the accumulation of food, saliva and necrotic matter. If these materials are not eliminated naturally, tonsilloliths develop, which pose a 10-fold increased risk of abnormal VSC levels [38]. On examination, the tonsils may or may not be hypertrophied and are usually not hyperemic. The microorganisms most often involved are streptococci, but viral infections are also a possible etiology (e.g., infectious mononucleosis). Anaerobic bacteria detected in tonsilloliths belonged to the species of *Eubacterium*, *Fusobacterium*, *Porphyromonas*, *Prevotella*, *Selenomonas* and *Tanerella*, all of which appear to be associated with the production of VSCs [39].

Bronchitis and pneumonia from aerobes, mycoplasma, or viruses, even if productive of a substantial amount of sputum, do not usually cause halitosis. Several clinical situations of the respiratory tract that present halitosis as a relevant symptom are described. Among them, anaerobic lung abscess, foreign bodies lodging along the respiratory tract, necrotizing pneumonia, emphysema, lung cancer, bronchiectasis and tuberculosis [40,41,42,43].

The lungs function as a source of odors that arise from metabolism. Aromatic foods such as garlic, alcohol, high-fat diets and ketosis (diabetic ketoacidosis), as well as nitrates, chloral hydrate and iodine-containing drugs, are examples of products that reach the lungs through the circulation and are released through expired air [36].

##### Gastrointestinal

Although the medical community and many patients believe that halitosis mostly originates in the stomach, it is known that only 0.5% of halitosis cases originate from thegastrointestinal tract.There are, however, gastrointestinal pathologies that can be the cause of bad breath, including esophageal reflux, achalasia, pyloric stenosis or hiatal hernia. In these pathologies, there is a weakening or inhibition of esophageal closure, incomplete emptying into the stomach, resulting in retention of food, liquid, and saliva, causing oralmalodor [10,31,44]. 

On the other hand, gastroesophageal reflux, an extremely common disorder, rarely causes halitosis. The most frequent symptom is heartburn, and the associated odor is like that of a simple belch, similar to the odor of the most recent meal [45].

Malabsorption syndromes, gastric carcinomas, and some enteric infections, bezoars have been noted to contribute to halitosis [45,46]. 

##### Metabolic Disorders 

Several metabolic diseases maymanifest in the form of bad oral odors, namely, DM, renal disease or liver cirrhosis [47,48]. Patients with type 2 DM (T2DM)manifest a typical sweet and fruity odor [49].

Using gas chromatography-mass spectrometry, it is possible to diagnose different extra-oral causes of halitosis, such as DM [50]. Diabetic ketoacidosis leads to a typical breath odor [51]. Patients with chronic renal failure have high blood urea nitrogen levels and reduced salivary flow. The odor is typically uremic associated with dry mouth. Treatment with peritoneal dialysis reduces the problem [47]. Feller and Blignaut described pancreatic insufficiency as a cause of extraoral halitosis as well [52]. 

Whittle et al., describe metabolic disorders in the intestines as a cause of halitosis. Trimethylaminuria, also known as “fish odour syndrome”, is a genetic disorder characterized by the body’s inability to metabolize trimethylamine. The accumulation of this volatile compound leads to its excretion in the urine, but it is also found in sweat and breath, giving it a fishy smell. According to the authors, this genetic disorder is the biggest cause of undiagnosed body odor, with social consequences, leading to isolation and even depression [53].

The symptoms of trimethylaminuria can be improved by changes in the diet to avoid precursors, in particular trimethylamineN-oxide which is found in high concentrations in marine fish. Treatment with antibiotics to control bacteria in the gut, or activated charcoal to sequester trimethylamine, may also be beneficial [54].

##### Hepatology and Endocrinology 

When liver function is reduced, waste products are eliminated through the lungs, causing ‘fetor hepticus’: a sweet, excremental odor (the breath of death) [55]. Liver failure inhibits detoxification throughout the body, causing unpleasant odors, called Fetor hepticus [56]. In addition, some hereditary disorders can influence breathing: tyrosinemia is the most important example (cabbage odor). Not only the hormonal cycle seems to influence the mouth odor, but also a lot of other intestinal diseases [57,58]. Recently, van Steenberge mentioned a completed list of metabolic and endocrinological aspects in correlation to halitosis: adult T2DM, pediatric Type1-DM, intestinal obstruction, alcoholic ketoacidosis, kidney-insufficiency, Trimethylaminuria, Phenylketonuria, Methionine adenosyl transferase deficiency, Isovaleria anacidity, Deficiency on chromosome 15, Maple syrup urine disease, Homocystinuria, Disease of Lignac [49].

##### Medication

In addition to medication resulting in a dry mouth that indirectly contribute to halitosis, such as anticholinergics, antihistaminic, antipsychotics, anxiolytics, antidepressant, antihypertensives, diuretics and narcotics; recently the use of bisphosphonates can contribute to oral malodor. Bisphosphonate, as potent drugs inhibiting bone resorption, is used in treatment of malignant bone tumors and their metastases [59]. In recent years, dentists and oral and maxillofacial surgeons reported continuously increasing cases of bisphosphonate-related osteonecrosis of the jaws (BRONJ). This disease is clinically characterized by exposed bones, formation of sequestrum, pain, and halitosis. Provided that pathogenesis of BRONJ is unclear, effective treatments for this disease are currently unavailable.The necrotic sequesters should be removed and it is tried to cover up the necrotic area with a steeled flap [60].

In a recent systematic review, medications that can cause extra-oral halitosis were identified: Aminothiols (cysteamine), Acid reducers (ranitidine), Anticholonergics (oxybutynin, glycopyrrolate), Antidepressants(imipramine, duloxetine), Antihistamines, steroids (astemizole, beclomethasone diproprionate), Anti-spasmodics (Colpermin), Chemoterapeutic agents (PX-12, sylibinphytosome), dietery supplements (fish oil, selenium, vitamin E), DMSO and Diclofenac. Further studies are needed to definitely determine the role of various medications in causing extra-oral halitosis [61].

#### 2.2.2. Intra-Oral Causes

Several review articles have shown that the oral cavity is the main contributor to bad breath in 85–90% of patients with halitosis [6,7,62]. The humid environment and temperature up to 37 °C inside the oral cavity favors the growth of bacteria and their ability to metabolize sulfur-containing amino acids (L-cysteine + L-methionine) to generate hydrogen sulfide and methyl mercaptan [21,29,63].

Oral malodor is mainly caused by microbial degradation of both sulfur-containing and non-sulfur-containing amino acids derived from proteins in exfoliated human epithelial cells and white blood cell debris, or present in plaque, saliva, blood, and tongue coating [6,64]. The subgingival periodontal biofilm is mainly composed of Gram-negative anaerobic bacterial species, which are proteolytic in nature [48]. 

These species are able to degrade sulfur-containing substrates on different surfaces of the oral cavity, including periodontal pockets, releasing volatile sulfur compounds (VSCs). Clinical studies have shown that VSCs are major contributors to halitosis [7,8]. Hydrogen sulfide, methyl mercaptan and, to a lesser extent, dimethyl sulfide, account for 90% of the VSC in bad breath [7,8,65].

##### Odontogenic Halitosis

Insufficient or inadequate oral hygiene, plaque, tooth decay, food impaction and poorly sanitized acrylic dentures (used at night or not regularly cleaned or with rough surfaces) are the main causes of odontogenic halitosis [66]. The dentist should be the first to diagnose these causes and follow up with the appropriate treatment for the cause.

##### Tongue Coating

Several studies have implicated the dorsum of the tongue as the main site of putrefaction of the microflora and the production of VSC [40], which is why it is considered the major contributor to intra-oral halitosis [40,67,68].

The dorsum of the tongue has a complex papillary structure that favors the accumulation of bacteria [69,70]. These microorganisms (e.g., *Veillonella* and *Actinomyces*), especially the Gram-negative and proteolytic nitrate-producing anaerobes, have the ability to produce odorous substances from food remnants and epithelial cell debris [4,5].

It is thought that instead of a few dominant species involved, interactions between various bacterial species occur on the dorsum of the tongue that result in oral malodor [4].

The score-based classification of tongue coating was developed by Kojima [71,72]. Clinically, there is a correlation between halitosis and tongue coating, and this association is particularly strong in the region posterior to the circumvallate papillae, an area with the highest load of Gram-negative bacteria that contribute to oral malodor [62].

This area is the most inaccessible to tongue hygiene procedures, hence the greatest accumulation of bacteria in this area.

The coating on the tongue is difficult to remove. Daily tongue scraping or brushing helps to reduce the substrate for putrefaction, bacterial load, and improve taste sensation [73].

There are several local factors, including salivary pH, reduced ambient oxygen concentration, bacterial production, and the substrate available for bacterial metabolism, that favor increases in salivary concentrations of VSC precursors, such as cystine and methionine. The production and release of these volatile compounds gives rise to the subsequent detection of these malodorous oral substances [57].

Decreased salivary flow; saliva stagnation; reduction of the carbohydrate content available as a bacterial substrate; and increased oral pH (the malodor occurs mainly in an alkaline microenvironment), create a favorable environment for the change from gram-positive to gram-negative bacteria [74].

##### Periodontal Disease

Periodontal diseases (gingivitis and periodontitis) are the oral inflammatory conditions that most often contribute to oral malodor, with the production of a very distinct, fetid or putrid smell [41]. Necrotizing gingivitis or periodontitis cause extreme soiled odors.

In patients with active periodontitis there is a significantly higher prevalence of damaged epithelial cells, leukocytes and bacteria in saliva compared to normal subjects [75].

Many studies support a direct correlation between periodontal disease and halitosis [67,76]. Other studies also verify a positive correlation between the depth of the pockets and the concentration of sulfur compounds [76,77].

The proposed microbiological link between halitosis and periodontal disease is based on three assumptions:(1)Periodontal patients have a higher prevalence of intraoral bacteria (bacterial plaque and tongue coating), decreased pH, which is necessary for amino acid putrefaction and formation of VSCs [78].(2)The microbially generated VSCs (hydrogen sulfide and methylmercaptan) facilitate the penetration of lipopolysaccharide into the gingival epithelium, inducing inflammation [79,80].(3)VSCs also aid in bacterial invasion of connective tissue by their toxic effects on epithelial cells, while methyl mercaptan prevents the growth and proliferation of epithelial cells [25].

This mechanism is enhanced by the decrease in oxygen tension due to the increase in the depth of the periodontal pocket, with a concomitant decrease in pH, which is necessary for the putrefaction of the amino acids that create VSCs [25].

However, some studies have not found a correlation between periodontal disease and halitosis, considering that the periodontal pocket is a closed environment, not representing a sufficient cause for the release of smelly gases that are able to escape into the mouth [21,79,81,82].

##### Oral Candidiasis

Long-term treatment with antibiotics or corticosteroids, immunosuppressed patients with HIV, DM, undergoing chemotherapy/radiotherapy, may develop fungal infections in the oral cavity. Oral candidiasis is a frequent infection, which presents several clinical manifestations in the oral mucosa.

In most cases of oral candidiasis, the diagnosis is based on clinical signs and symptoms through physical examination and medical history. Confirmation of the diagnosis can be made with complementary tests, especially when the clinical examination is uncertain, or the patient does not respond to antifungal therapy. Confirmation of the diagnosis can be made by obtaining a smear (exfoliative cytology). Candida infections produce a distinct sweet, fruity odor. Antifungal agents, such as nystatin, topical ketoconazole, clotrimazole and miconazole, can resolve the condition and treat oral malodor [40].

##### Oral Cancer

Since malignant or benign primary tumors of the oral cavity are very often associated with the presence of necrotic tissue, blood exudation, opportunistic infections, accumulation of food debris, an environment is created that favors increased putrefaction and VSC production.

Patients undergoing cancer treatment are more susceptible to tissue destruction, multiple infections and bleeding. All these factors contribute to the accumulation of anaerobic bacteria and the release of foul-smelling gases [83,84].

##### Other Oral Sources

There are other factors that contribute to intraoral halitosis which include: non- vital teeth, exposed tooth pulps, healing wounds, stomatitis, intra-oral neoplasia, extraction wounds (with blood cloth or purulent discharges) or crowding of teeth (favoring food entrapment) canal sob involved and fixed orthodontic appliances. Moreover, peri-implantitis, pericoronitis, recurrent oral ulcerations and herpetic gingivitis, are described as an origin for bad breath [85,86].

All these factors create a food retention site or plaque that allows for bacterial putrefaction of amino acids, causing halitosis [66]. Acute clinical situations such as pericoronal infections, oral ulcerations and necrotizing ulcerative gingivitis can also cause oral malodor with the production of a characteristic oral stench. There is also temporary halitosis that lasts only a few hours, caused for example by eating foods that contain VSCs, such as garlic or fast food [87]. On the other hand, a high fiber diet such as vegetables, fruit and green tea accelerate gastric emptying, leading to reduced VSC levels over a period of time [88,89].

## 3. Treatment

Apatient who suffers from halitosis and seeks help is a person who is always worried about it, who has often tried treatments and has not found an answer [90]. An accurate objective diagnosis must be made in order to manage the appropriate treatment for the causal factors. All patients diagnosed with halitosis, regardless of the cause of halitosis, should be evaluated by an oral health professional.

The authors describe a treatment approach based on halitosis types in Figure 3.

### 3.1. Treatments for Intra-Oral Causes

The course of treatment for halitosis is determined after a thorough oral clinical examination, including the dentition, soft tissue, and periodontal health status. All active caries, secondary caries, pulp pathology, oral pathologies such as chronic ulcerative conditions, oral candidiasis and xerostomia, must be identified, diagnosed and treated appropriately. Moreover, the diagnosis of periodontal diseases such as gingivitis, periodontitis or necrotizing periodontal diseases, should be performed and adequately treated, since these pathologies are a major contributor to the levels of oral VSC [79].The treatment of intra-oral halitosis include four phases: (1) mechanical reduction of nutrients and intraoral microorganisms; (2) chemical reduction of microorganisms; (3) inversion of volatile gases into non-volatile components (chemical neutralization of VSC) or (4) masking the malodor [91].

#### 3.1.1. Mechanical Reduction

Bearing in mind that microorganisms and their metabolites are involved in halitosis etiopathogenesis, mechanical removal of biofilm and microorganisms is the first step in halitosis control [92]. Tongue coating is the main causal factor of intraoral halitosis, hence the importance of extensive cleaning of the tongue. Scraping the dorsum of the tongue reduces both available nutrients and available microorganisms, leading to improved odor [93]. All patients should receive clear instructions on the most appropriate oral hygiene care for their case, as well as an explanation of tongue cleaning. The patient should gently brush the dorsum of the tongue with a soft bristle brush and a toothpaste in 5 to 15 movements. The area that tends to accumulate bacterial deposits and keratin and food debris, contributing to physiological halitosis is terminal sulcus, is the division between the posterior and middle thirds of the tongue. Removal of these materials decreases the release of VSCs.

There are two ways to do daily tongue cleaning: using a regular toothbrush or using atongue scraper [94]. It was described in a systematic review published by Van der Sleen et al. that tongue brushing or tongue scraping allows the reduction of tongue coating and improvement of halitosis. According to these authors, tongue scrapers are suitable for the anatomy of the tongue and reduce 75% of VSCs compared to 45% when a toothbrush is used [95].

On the other hand, in a Cochrane review in 2006, which compared randomized controlled trials for different tongue cleaning methods, concluded that there was a weak, but statistically significant, difference in the reduction of VSC levels when scrapers or cleaners were used instead of toothbrushes [96]. More studies are needed for clearer conclusions.

Since periodontitis is a major cause of oral malodor, treating periodontitis will also improve halitosis.

One-stage full-mouth disinfection can be performed, as described by Bollenet et al. [97], combining scaling and root planing with the use of chlorhexidine. There is a significant microbiological reduction up to 2 months, with reduced organoleptic scores [98].

The accumulation of bacterial plaque due to lack of interproximal cleaning leads to a high incidence of malodor, so it is essential to use dental floss/interproximal brush to control bacterial plaque and oral microorganisms [99].

The oral health professional should dedicate time from their consultation to the motivation and instruction of oral hygiene care to patients. This is the only way to achieve good treatment adherence results [100].

#### 3.1.2. Chemical Reduction

Antibacterial agents for mouthwashes include chlorhexidine (CHX), cetylpyridinium chloride (CPC), and triclosan. Its mechanism acts on bacteria capable of producing volatile sulfur compounds [29].

Mouthwashes containing CHX and CPC can inhibit the production of VSCs, while mouthwashes containing chlorine and zinc dioxide have a neutralizing action on sulfur compounds that produce halitosis, according to a Cochrane review [101].

Rinsing is a common practice in the management of oral malodor. The most used rinse components are:Chlorhexidine (CHX): considered the gold standard mouth rinse for halitosis treatment [102]. Its use at a concentration of 0.2% causes a 43% reduction in VSCs and a 50% reduction in organoleptic scores throughout the day [99]. CHX in combination with CPC produce greater reductions in VSCs level, and both aerobic and anaerobic bacterial counts showed the lowest percentage of survival in a randomized, double-blind, cross-over study design [102]. Combined effects of zinc and CHX were studied in a study conducted in 10 participants, Zinc (0.3%) and CHX (0.025%) in low concentration led to 0.16% drop in H_2_S levels after 1 h, 0.4% drop after 2 h and 0.75% drop after 3 h showing a synergistic effect of the two [103]. However, patients may be reluctant to use CHX long-term as it has an unpleasant taste and can cause (reversible) staining of the teeth [104].Essential oils: these products give only a short-term and restricted effect (25% reduction) for 3 h. Furthermore, the reduction in odor-producing bacteria is limited [105]. Usage of Listerine containing essential oils resulted in significant reduction in halitosis-producing bacteria in healthy subjects [106].Chlordioxide: It is a strong oxidant that can reduce halitosis by oxidizing H_2_S, CH_3_SH, cysteine and methionine. A 29% reduction in odor was reported after 4 h [107].Triclosan, is a widely used antimicrobial agent with good results in reducing dental plaque, gingivitis and halitosis [108]. Its use in toothpastes in combination with a tongue scraper and toothbrush revealed a significant reduction in organoleptic scores and sulfur levels in the mouth air [108].A formulation of triclosan/copolymer/sodium fluoride in 3 weeks randomized double blind trial by Hu et al. seemed to be particularly effective in reducing VSC, oral bacteria, and halitosis [109].

Toothpastes containing stannous fluoride, zinc or triclosan have a beneficial effect on reducing oral malodor for a limited period of time [110,111,112].

In a recent Cochrane review by Fedorowicz, with five randomized controlled trials involving 293 participants, 0.05% chlorhexidine + 0.05% cetylpyridinium chloride + 0.14% zinc lactate was compared to placebo. With the use of this mouthwash, there was a significant reduction in organoleptic scores, but also a more significant presence of stains on the tongue and teeth. A meta-analysis of the data was not possible due to clinical heterogeneity between trials [101].

It is concluded that this mouthrinse (0.05% chlorhexidine + 0.05% cetylpyridinium chloride + 0.14% zinc lactate)plays an important role in reducing the levels of halitosis producing bacteria on the tongue and can be effective in neutralization of odoriferous sulfur compounds. However, well-designed, randomized controlled trials with larger sample size, a longer intervention and follow-up period are still needed to confirm these results.

#### 3.1.3. Probiotics

Several studies have shown that probiotic bacterial strains, originating from the indigenous oral microbiota of healthy humans, may have potential application as adjuvants for the prevention and treatment of halitosis [113]. The aim is to prevent the re-establishment of unwanted bacteria and thus limit the recurrence of oral malodor for an extended period. Recently, several studies have been carried out to replace the bacteria responsible for halitosis with probiotics such as *Streptococcus salivarius* (K12), *Lactobacillus salivarius* or *Weissellacibaria*.

Oral administration of probiotic lactobacilli has shown good results in the treatment of physiological halitosis, as well as improved bleeding on probing of periodontal pockets [114].

Furthermore, in vivo and in vitro studies revealed that *Weisellacibaria* isolates have the ability to inhibit the production of VSC, demonstrating that they have potential for the development of new probiotics for use in the oral cavity [115]. The use of a suspension of living non-pathogenic *Escherichia coli* bacteria also seems to have good results in the treatment of gut-caused halitosis [116].

#### 3.1.4. Transformation of Volatile Sulfur Components

Products containing chlorite anion and chlorine dioxide have been shown to be effective in oxidizing and inactivating the oral VSC demonstrating long-lasting effects [117,118]. Positively charged metal ions such as zinc, mercury and copper bind to sulfur radicals inhibiting the expression of VSCs [119,120]. For this reason, the combination of zinc and CHX appears to have a synergistic effect on the elimination of VSCs. According to the authors Young et al., a commercial rinse (containing 0.005% CHX, 0.05% cetylpyridinium chloride (CPC) and 0.14% zinc lactate) appears to be much more efficient than CHX alone, due to the zinc effect [119].

#### 3.1.5. Masking Effect

Rinsing products, sprays, mint tablet, chewing gum increase the saliva production, thereby retaining more soluble sulfur components for a short period of time, having only a short-term masking effect [121].

### 3.2. Treatments for Extra-Oral Causes

Bearing in mind that halitosis presents a multifactorial complexity, treatment should be individualized and directed to each patient, rather than generalized [1].

Diagnosis and treatment involve a multidisciplinary team: primary healthcare clinician, dentist, otolaryngologist, nutritionist, gastroenterologist and clinical psychologist [122].

After a detailed clinical oral examination and anamnesis, that excludes intra-oral causes for halitosis, patients with signs or symptoms of systemic diseases that may be the cause of oral malodor, should be referred to the appropriate medical specialty (ENT, pulmonologist, endocrinologist or gastroenterologist). Patients with pseudohalitosis or halitophobia should be counseled appropriately and referred for psychologic evaluation and treatment.

In the clinical approach to halitosis, a relationship of trust and empathy between the patient and a general practitioner is extremely important. In this reliable medical approach, the patient will feel comfortable to communicate their complaints and the doctor will be able to encourage the patient to undergo treatment, improving the quality of life of the patient as a whole, and improving their interactions and social relationships [104].

#### 3.2.1. Halitophobia

Imagined halitosis is poorly documented in the psychiatric literature [123]. Many of the cases with imagined halitosis described in the literature resemble the psychiatric syndrome of social phobia [124]. Generally, these patients believe that their oral malodor is related to social rejection or avoidance behaviors of the people with whom they interact [125]. Patients with halitophobia require referral for clinical psychology investigation and treatment for mental assessment and appropriate treatment [104,122].

The ‘treatment’ of these patients is impossible, as they are not within the arguments presented by a physician. Mostly, these patients hop from clinic/specialist to clinic/specialist to find an argument for their self-reported problem.

#### 3.2.2. Dry Mouth/Xerostomia

It is important to accurately describe and differentiate dry mouth problems. The subjective feeling of dry mouth is defined as xerostomia, while hyposalivation is the objective finding of decreased salivary production [126].

The treatment of hyposalivation or dry mouth will also contribute to the treatment of halitosis. The dry mouth symptom can be treated with hydration and sialogogues or with artificial saliva substitutes [126].

When the cause of dry mouth is medication, it becomes important to find other pharmacological alternatives without compromising the patient’s health.

The patient should be encouraged to increase water intake and avoid drinking caffeinated beverages.

When there is a complaint of dry mouth sensation—Xerostomia—for salivary stimulation, we can resort to the use of sugar-free candies or gums and also the use of an artificial salivary substitute, which is usually composed of carboxymethylcellulose.

In severe cases of dry mouth, e.g., patients with Sjögren’s syndrome or patients undergoing radiotherapy, therapy with a cholinergic agonist is prescribed. The most frequently used has been pilocarpine at a dosage of 5 to 10 mg/day [1], and more recently cevimeline hydrochloride (Evoxac), 30 mg three times a day, also with good results in the treatment of dry mouth [126,127].

## 4. Conclusions

Halitosis is highly prevalent with multifactorial origins, and high burden for social and self-esteem. This review emphasizes the importance for a multidisciplinary approach. Despite thecurrentdecision trees for the clinical management of halitosis, there is still some inconsistencies that requirerobust randomized clinical trials comparing standard and innovative therapies.

## Figures and Tables

**Figure 1 ijerph-19-11303-f001:**
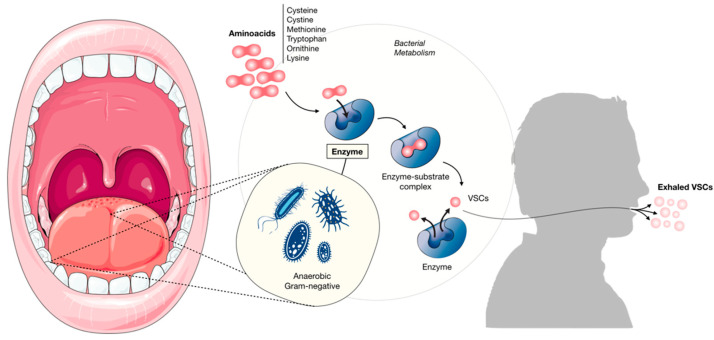
Pathophysiology of halitosis. Bacteria present in the gingival pockets and the dorsum of the tongue metabolize amino acids through enzymes into volatile sulfur compounds that are exhaled.

**Figure 2 ijerph-19-11303-f002:**
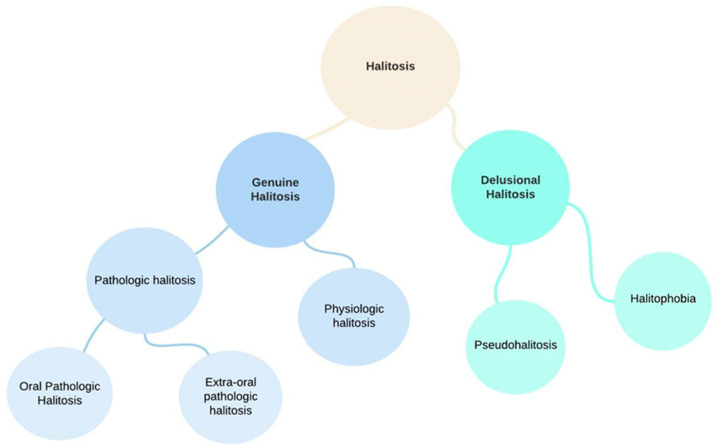
Current classification of halitosis.

**Figure 3 ijerph-19-11303-f003:**
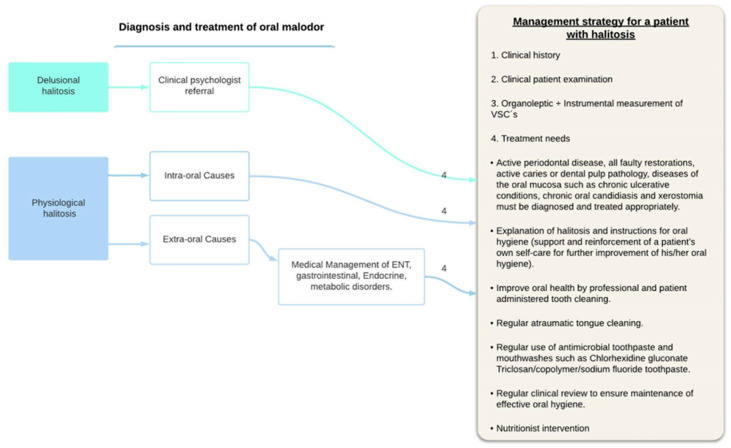
Treatment approach based on halitosis types and etiology.

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
