# Peer review of "Revisiting Standard and Novel Therapeutic Approaches in Halitosis: A Review"

_ijerph, 2022, doi:10.3390/ijerph191811303_

Round 1

Reviewer 1 Report

While the paper is interesting and well written and fits the aims of the Journal I am concerned over three major issues:

1. While bad breath or the so called ‘halitosis’ is oral health problem that can affect one’s self-esteem and social relations, the Authors should also refer, at least in the Introduction, to the concept of medicalization and the so called ‘disease mongering’. See for example:

-- Dossey L. Listerine's long shadow: disease mongering and the selling of sickness. Explore (NY). 2006 Sep-Oct;2(5):379-85. doi: 10.1016/j.explore.2006.06.005. 

-- Conrad P, Leiter V. From Lydia Pinkham to Queen Levitra: direct-to-consumer advertising and medicalisation. Sociol Health Illn. 2008 Sep;30(6):825-38. doi: 10.1111/j.1467-9566.2008.01092.x.

-- Lynn Payer Disease-Mongers: How Doctors, Drug Companies, and Insurers Are Making You Feel Sick 1992.

2. More importantly, writing a review of literature is not a description writing, it is an outline of research. Thus, although the paper aims to provide the review of research on halitosis’ etiology, classification, and treatments it lacks the methods section describing a detailed flow of information through the different phases of the reviews and map out the number of records identified, included and excluded. Thus, while there is no research question the review intends to address, neither the inclusion/exclusion criteria are described nor the exact number of articles analysed is given. Consequently, the search strategy should be elaborated more deeply. For example, the PRISMA statement could be applied and the results of the identification, screening, eligibility assessment and final inclusion could be presented in PRISMA 2009 Flow Diagram (Moher, D.; Liberati, A.; Tetzlaff, J.; Altman, D.G.; The PRISMA Group. Preferred Reporting Items for Systematic Reviews and Meta-Analyses: The PRISMA Statement. PLoS Med. 2009, 6(7), e1000097. doi:10.1371/journal.pmed1000097). All in all, 1. the inclusion/exclusion criteria should be explained and justified, and 2. each of the steps included in the PRISMA procedure should be briefly described: the identification, screening, eligibility assessment and final inclusion.

3. As it supposed to be a review paper the analysis of existing research should be aborted more deeply.

To conclude, while the topic fits well with the aims of Journal and the issues raised in this paper are important and timely, I believe that in its current form it cannot be published. However, after some revision it may be considered for another evaluation.

Reviewer 2 Report

Please see comments in uploaded file

Round 2

Reviewer 1 Report

After reading the Authors’ response and the revised manuscript itself I believe that the Authors have clarified most of the issues raised in the review and that this revised manuscript is now more consistent owing to their corrections and additional arguments. On the whole, I appreciate this effort and have no further concern regarding the manuscript.

Reviewer 2 Report

See uploaded file in response to author response
